

# Association of late-onset postpartum depression of mothers with expressive language development during infancy and early childhood: the HBC study

Sona-Sanae Aoyagi[1,2], Nori Takei[1,3], Tomoko Nishimura[1,3], Yoko Nomura[3,4] and Kenji J. Tsuchiya[1,3]

[1] United Graduate School of Child Development, Hamamatsu University School of Medicine, Hamamatsu, Japan
[2] School of Education, Meisei University, Tokyo, Japan
[3] Center for Child Mental Development, Hamamatsu University School of Medicine, Hamamatsu, Japan
[4] Queens College and Graduate Center, City University of New York, NY, United States of America

Corresponding author
Kenji J. Tsuchiya,
tsuchiya@hama-med.ac.jp

## ABSTRACT

**Background.** While it has been implied that an infant's exposure to maternal postpartum depression (PPD) may be associated with delayed development of expressive language, it remains unclear whether such a delay persists into childhood and whether the onset of PPD onset—early (within 4 weeks after childbirth) vs. late (between 5 and 12 weeks postpartum)—is relevant in this context.

**Objective.** To examine whether children of mothers with early- or late-onset PPD have reduced expressive language scores during infancy and early childhood (up to 40 months of age).

**Methods.** This longitudinal, observational study was conducted as a part of the Hamamatsu Birth Cohort for Mothers and Children (HBC Study), a population-representative sample in Japan. A total of 969 neonates and their mothers were included in the analysis.

**Exposures.** Early- and late-onset PPD was measured using the Edinburgh Postnatal Depression Scale.

**Main Outcomes and Measures.** Expressive language development was measured using the Mullen Scales of Early Learning. Six points over time were monitored (10, 14, 18, 24, 32, and 40 months postpartum). The relationship between the exposure variable and any change in expressive language score was evaluated using multiple linear regression analysis and growth curve analysis, both adjusted for covariates.

**Results.** Results from the adjusted regression analysis showed that children of mothers with late-onset PPD had significantly lower expressive language scores at 18 months of age and beyond, with a score reduction of approximately 0.6 standard deviations from the reference value at 40 months of age (95% CI [−0.888 to −0.265], $p < .001$). This association was confirmed on growth curve analysis, which revealed a significant, monotonic decline of expressive language development between 10 and 40 months of age among children of mothers with late-onset PPD, but not among children of mothers with early-onset PPD.

**Conclusion.** Exposure to late-onset PPD may lead to a persistent decline in the rate of expressive language development in offspring during infancy and early childhood,

highlighting the significance of monitoring for late-onset PPD to facilitate early detection and intervention.

# INTRODUCTION

Pregnancy and the postpartum period are associated with physical and emotional changes for the mother. These changes, including hormonal withdrawal (*Harris et al., 1994*), can cause depressive symptoms among parturients ranging in form from mild to severe (*Pearlstein et al., 2009*). Literature has indicated that a substantial proportion of women exhibit postpartum depressive symptoms (also referred to as postpartum depression; PPD) approximately three months after childbirth (*O'Hara, 2009*; *Stewart & Vigod, 2016*). Maternal PPD is clinically evaluated according to measurement scales for screening purposes; the Edinburgh Postnatal Depression Scale (EPDS) is one of the most frequently adopted scales in the literature (*Chaudron et al., 2010*; *Matthey, 2008*). Using the EPDS, both systematic and expert review studies conclude that the prevalence of PPD ranges from approximately 10 to 20% (*Pearlstein et al., 2009*; *Shorey et al., 2018*; *Matsumoto et al., 2011*). Although previous work has documented the high prevalence of PPD and its potentially serious outcomes among parturients (including suicide; *Rodriguez-Cabezas & Clark, 2018*), PPD has remained a major public health concern worldwide. Thus, specialists have recommended that parturients should be more closely observed during their first year postpartum (*Van der Zee-van den Berg et al., 2017*).

Apart from maternal outcomes, one major concern of PPD is that it may compromise neurodevelopment in the offspring—particularly their language skills (*Keim et al., 2011*; *Quevedo et al., 2011*). However, such delay in language is typically observed during early infancy and has yet to be examined closely after the child's first year (*Murray, 1992*; *Koutra et al., 2013*; *Smith-Nielsen et al., 2016*). On the other hand, language delay in young children is associated with delays in school readiness and behavioral problems in later years (*Benner, Nelson & Epstein, 2002*; *High, 2008*). Of note, there are limited longitudinal studies investigating the trajectories of language development among children born from mothers with PPD (*Kingston, Tough & Whitfield, 2012*).

One possible underlying mechanism in the relation between maternal PPD and infant language outcomes is that PPD has an adverse effect on the mother-child relationship (e.g., insecure attachment; *Murray, 1992*; *Cooper & Murray, 1998*; *Tomlinson, Cooper & Murray, 2005*). Indeed, the infant's health and psychological well-being through lifetime is dependent in part on the physical and psychosocial care that they receive from their mothers in the early life (*Birkeland, Breivik & Wold, 2014*; *Ohoka et al., 2014*; *Kerns & Brumariu, 2014*; *Bosmans & Kerns, 2015*). As such, some authors have suggested that the association between maternal PPD and infant language outcomes is due to decreased maternal responsiveness to the child (*Keim et al., 2011*; *Conroy et al., 2012*; *Kingston,*

*Tough & Whitfield, 2012*). However, other studies have reported no such relationship (*Murray, 1992*; *Cornish et al., 2005*; *Koutra et al., 2013*). The discrepancies could be due to methodological limitations, including small sample size (*Sharp et al., 1995*; *Murray et al., 1996a*; *Quevedo et al., 2011*; *Azak, 2012*; *Smith-Nielsen et al., 2016*), retrospective design, and non-representative sample selection leading to biased estimates (*Murray, 1992*; *Murray et al., 1996b*).

One potential protective agent against the effects of maternal PPD on infant language development is breastfeeding. It is well documented that breastfeeding has a beneficial effect on a child's cognitive and motor development (*Dee et al., 2007*; *Leventakou et al., 2015*). Any examination on the association between maternal PPD and infant language development should be controlled for duration of breastfeeding.

Another confounding variable in the association between maternal PPD and infant language outcomes stem from the heterogeneity of PPD (*Mori et al., 2011*; *Kettunen, Koistinen & Hintikka, 2014*). Specifically, the time window during which PPD had been examined in previous studies is inconsistent. The American Psychiatric Association definition of postpartum onset is restricted to depressive episodes beginning within 4 weeks postpartum. However, a number of studies have indicated that the postpartum period in which depressive symptoms are more likely to occur, compared with other periods, can last as long as three months (*Mori et al., 2011*; *Kettunen, Koistinen & Hintikka, 2014*). In this context, the division of PPD into two types (i.e., early-onset PPD: occurring within four weeks postpartum; and late-onset PPD: occurring during the fifth to twelfth week) is justifiable in terms of predictors and outcomes (*Mori et al., 2011*; *Terp & Mortensen, 1998*; *Munk-Olsen et al., 2006*). Furthermore, it has been well documented that early-onset PPD does not last long. To clarify whether maternal PPD exerts any influence on the infant's neurodevelopmental growth, it is necessary to conduct a longitudinal study of language development indices with a special attention on the timing of the occurrence of PPD (i.e., early-onset PPD vs. late-onset PPD). In the current study, we used a representative birth cohort to investigate whether early- and late-onset PPD are associated with reduced expressive language development during infancy and early childhood at 10, 14, 18, 24, 32, and 40 months of age (*Takagai et al., 2016*).

## MATERIALS & METHODS

### Participants

The present investigation was conducted as part of the ongoing Hamamatsu Birth Cohort for Mothers and Children (HBC Study), which is described in detail elsewhere (*Tsuchiya et al., 2010*; *Takagai et al., 2016*). Briefly, the HBC Study enrolled a consecutive series of mothers ($n = 1{,}138$) and their infants ($n = 1{,}258$) born between 24 December 2007 and 9 March 2012. It included offspring (i.e., siblings) who were born from the same mother during this entry period. All women who visited either of the two research sites (Hamamatsu University Hospital and Kato Maternity Clinic) during their first or second trimester of pregnancy were invited to participate. There was no difference in demographic characteristics between the two sites. All mothers who agreed to participate in the study

(including those recruited from Kato Maternity Clinic) gave birth at the Hamamatsu University Hospital, where all postpartum assessments were performed. Based on the reports from the Department of Health, Labor and Welfare in Japan (*MHLW, 2013*), we confirmed that the enrolled cohort was representative of the general population of Japanese mothers (age, socioeconomic status, and parity) and their offspring (birthweight and gestational age at birth; *Takagai et al., 2016*). The present analysis included the mother and child dyads with at least two PPD evaluations witihin the first three months postpartum.

## Measurements

### Outcome variable: expressive language development

Infant neurodevelopment was evaluated in terms of expressive language development, using the Mullen Scales of Early Learning (MSEL; *Mullen, 1995*). The MSEL is a one of the child composite scales, such as the Bayley Scale of Infant Development (*Bayley, 1993*); it was developed to measure a range of aspects of neurodevelopment in children from birth through 69 months of age. The MSEL provides an overall assessment in neurodevelopment, and covers the following five domains: Expressive Language, Receptive Language, Visual Reception, Gross Motor, and Fine Motor. Among these domains, we used the score of the Expressive Language domain in this study, which reflects children's spontaneous utterances, specific vocal or verbal responses to tasks, and high-level concept formation. The Expressive Language scale consists of 28 items with the total score ranging from 0 to 50, and is organized in a developmentally hierarchical manner; moreover, developmental stages from 1 to 8 were set accordingly. For instance, at least three consonants (e.g., "m", "p", "d", "k" sounds) and two syllables (e.g., "mama", "papa") may or may not be observed at ca. 10 months (stage 3), although such early language expressions are quite likely to be observed at later stages, and therefore they are assessed as score 1. Similarly, communicative expression using at least one word other than "mama" and "papa" may or may not be observed at 14 months (stage 4), although such word expressions are likely to be observed at later stages in later years, and therefore they are assessed as score 1. Accordingly, participating children were assessed at 10, 14, 18, 24, 32, and 40 months of age by our research staff that were blinded to any kinds of research theme and to information about our cohort of mothers during the postpartum period. As language development is more expressive around 1 year of age, we used data from six of the nine time points (10, 14, 18, 24, 32, 40 months of age) to evaluate the child's developmental status. To facilitate age-specific analyses, the original expressive language scores from the MSEL were converted to $Z$-scores with reference to all of the data in our sample.

### Exposure variable: PPD

We evaluated PPD using the Japanese version of the EPDS, which is a 10-item self-report questionnaire that has been validated for use during the postpartum period (*Cox, Holden & Sagovsky, 1987*; *Murray & Carothers, 1990*). Mothers were asked to complete the EPDS questionnaire at three time points (around 2, 4, and 5–12 weeks postpartum) and return it to our research center by mail. Mothers were not given any incentives to complete the EPDS questionnaires. The EPDS scores were dichotomized with a cut-off value of 9, which is the diagnostic score validated by other studies conducted in Japan (*Tamaki, Murata*

*& Okano, 1997*; *Yamashita et al., 2000*; *Yoshida et al., 2001*). Three PPD categories were defined: early-onset PPD (defined as at least one EPDS ≥ 9 within 4 weeks postpartum); late-onset PPD (defined as at least one EPDS ≥ 9 during 5–12 weeks postpartum but not within 4 weeks); and no PPD (defined as no EPDS ≥ 9; *Mori et al., 2011*).

### Covariates

We considered the following covariates because they have been previously reported to be associated with either PPD or a delay in neurodevelopment of the infant: infant sex, birth order, twin birth, birthweight, gestational age at birth, duration of breastfeeding, maternal and paternal age, annual household income at infant's birth, maternal and paternal education level (*Sharp et al., 1995*; *Dennis & McQueen, 2007*; *Emond et al., 2007*; *Marques et al., 2015*), and maternal history of mood/anxiety disorders, as defined in the Diagnostic and Statistical Manual of Mental Disorders, 4th edition (*American Psychiatric Association, 2000*).

### Analysis

Assuming that approximately 10% of data points would be missing for each time point (10, 14, 18, 24, 32, 40 months) as was provisionally reported by our technical staff, we first tested whether the missing data were random using Little's Test (*Little, 1988*). Then, we applied multiple imputation methods according to the corresponding guidelines (*Graham, Olchowski & Gilreath, 2007*), setting the number of imputations at 20, using other developmental indices including age, sex, anthropological measures, as well as motor skills of the child. After this, we conducted multiple linear regression analyses to obtain estimates of sample-specific means. Specifically, the outcome measurements (i.e., expressive language scores at 10, 14, 18, 24, 32 and 40 months) were regressed on the PPD status variable (i.e., no PPD, early-onset PPD, late-onset PPD) without adjustment (crude model). Afterwards the model was adjusted for the covariate of maternal history of mood/anxiety disorders (model 1), which is recognized as the most influential risk factor for PPD (*Mori et al., 2011*; *Silverman et al., 2017*). Finally, the model was adjusted for all other covariates as mentioned above (model 2).

Growth curve analysis was conducted to detect PPD-related changes in expressive language development (sequential, repeated-measured outcome). The slope of the growth curve is the change in $Z$-score of expressive language score per month (i.e., growth from 10 months to 40 months of age), while the intercept corresponds to the mean estimated score at 10 months of age (*Rothman, 1990*). PPD status (i.e., early-onset PPD vs. late-onset PPD vs. no PPD) was entered as fixed parts, with interactions between maternal PPD status and a time point (months of age) or time point$^2$ (square of time point) tested simultaneously. Statistically significant interaction terms were retained in the model. Growth curve modelling was adjusted for all covariates entered into model 2. A random slope along the time points was also allowed for, with an estimation of unstructured covariance matrices. Clustering was also allowed because some infants were from the same mother. All analyses were conducted using Stata version 15.1 (StataCorp, College Station, TX, USA).

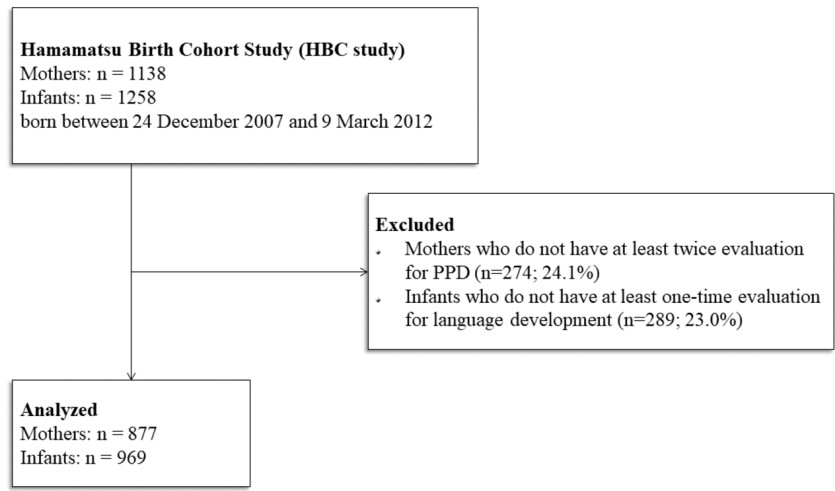

**Figure 1**  **Flow-chart of study participants.**

## Ethical considerations

The study protocol was approved by the Ethics Committees of Hamamatsu University School of Medicine and Hamamatsu University Hospital (20-82, 21-114, 22-29, 24-67, 24-237, 25-143, 25-283, E14-062, E14-062-1, E14-062-2, E14-062-3, 17-037). Written informed consent was obtained from each mother regarding her own and her infant's participation. The present study conforms to the STrengthening the Reporting of OBservational studies in Epidemiology (STROBE) statement.

## RESULTS

Figure 1 shows the flow of participants through the study. We excluded 23% of the original cohort (289 infants of 274 mothers), due to either the mother missing more than one EPDS assessment or the infant missing all follow-up evaluations between 10 and 40 months of age. The major reason for missing follow-up evaluations was conformance with the "satogaeri bunben" tradition ($n = 111$), a childbirth custom prevalent in Japan and some East Asian countries in which mothers temporarily move to their own mother's residence during the perinatal period (*Yoshida et al., 2001*; *Nishimura et al., 2016*).

We found no significant difference in socio-demographics (e.g., sex or birth order) between infants included in the analyses ($n = 969$; Table 1) to those excluded from the analysis ($n = 289$). However, the mother's age and level of education at childbirth were significantly higher among infants included in the analysis compared to those excluded (31.7 vs. 30.5 years, $t(1,256) = 3.73, p < .001$; 13.9 vs. 13.4 years, $t(1,256) = 4.52, p < .001$; respectively). The response rates for the EPDS questionnaire were very high at all three time points (99.8% at 2 weeks, 100% at 4 weeks and 82.8% at 5–12 weeks, respectively).

Fewer than 10% of data points were missing from each time point (10, 14, 18, 24, 32, and 40 months of age). Missing data for the language scores occurred completely at random ($p = .08$; *Little, 1988*). An application of the multiple imputations was justified, and 969

**Table 1** **Characteristics of the children (*n* = 969) and mothers (*n* = 877) included in HBC study.** Only six non-Japanese children were included in this cohort.

|  | *n* (%) or Mean (sd) | Range |
|---|---|---|
| Age in month of the child[a] |  |  |
| At 10 months (*n* = 936) | 10.5 (0.5) | 9.2–13.4 |
| At 14 months (*n* = 879) | 14.5 (0.6) | 12.1–20.6 |
| At 18 months (*n* = 924) | 18.6 (0.7) | 16.6–23.2 |
| At 24 months (*n* = 912) | 24.7 (0.9) | 21.0–33.8 |
| At 32 months (*n* = 883) | 33.2 (2.0) | 28.8–45.4 |
| At 40 months (*n* = 884) | 39.8 (2.2) | 32.8–59.6 |
| Infant sex |  |  |
| Male | 491 (51%) |  |
| Birth order |  |  |
| First-born | 485 (51%) |  |
| Second-born | 363 (37%) |  |
| Third-born or later | 121 (12%) |  |
| Multiparity |  |  |
| Twin birth | 30 (3%) |  |
| Birthweight (g) | 2,944 (436) | 946–4,286 |
| Gestational age at birth (in weeks) | 39.0 (1.5) | 30.1–42.1 |
| Duration of breastfeeding (in months) | 9.8 (6.2) | 0–23.1 |
| Maternal history of mood disorders | 99 (10%) |  |
| Maternal history of anxiety disorders | 35 (4%) |  |
| Maternal age at the infant's birth (in years) | 31.7 (5.0) | 17.7–44.6 |
| Maternal education level (in years) | 13.9 (1.9) | 9–23 |
| Paternal age at the infant's birth (in years) | 33.6 (5.8) | 18.9–53.4 |
| Paternal education level (in years) | 14.3 (2.6) | 9–26 |
| Annual household income at birth (million yen) | 6.2 (2.8) | 1.0–27.0 |
| Edinburgh postnatal depression scale |  |  |
| Score at 1st measurement (at the 2nd week) | 3.9 (3.6) | 0–22 |
| Score at 2nd measurement (at the 4th week) | 3.1 (3.4) | 0–21 |
| Score at 3rd measurement (between 5th and 12th weeks) | 2.6 (3.2) | 0–28 |
| Postpartum depression |  |  |
| No PPD | 823 (85%) |  |
| Early-onset PPD | 103 (11%) |  |
| Late-onset PPD | 43 (4%) |  |
| Expressive Language score[a,b] |  |  |
| At 10 months (*n* = 936) | −0.19 (0.96) | −2.58–2.86 |
| At 14 months (*n* = 879) | −0.09 (1.04) | −3.00–2.04 |
| At 18 months (*n* = 924) | −0.14 (0.95) | −3.00–2.66 |
| At 24 months (*n* = 912) | −0.05 (0.99) | −3.00–3.00 |
| At 32 months (*n* = 883) | −0.07 (0.99) | −3.00–3.00 |
| At 40 months (*n* = 884) | 0.00 (1.02) | −3.00–2.85 |

Notes.
[a]Not all participants attended all follow-up visits planned.
[b]$Z$-scores for which the mean is 0 and standard deviation (sd) is 1.

**Table 2  PPD-associated differences in expressive language scores.** All differences of regression coefficient are given with 95% confidence intervals. Scores obtained in the no PPD group were considered as reference. The results of unadjusted (crude) and adjusted models are given. Model 1 was adjusted for maternal history of depression or mood/anxiety disorders. Model 2 was further adjusted for infant sex, birth order, twin birth, birthweight, gestational age at birth (in weeks), duration of breastfeeding (in months), maternal and paternal age at the infants birth (in years), annual household income, and maternal and paternal education level (in years).

| | Crude | Model 1 | Model 2 |
|---|---|---|---|
| **At 10 months** | | | |
| Early-onset PPD | 0.094 (−0.108, 0.296) | 0.110 (−0.096, 0.316) | 0.088 (−0.117, 0.293) |
| Late-onset PPD | −0.058 (−0.360, 0.245) | −0.053 (−0.356, 0.250) | 0.014 (−0.287, 0.314) |
| **At 14 months** | | | |
| Early-onset PPD | −0.035 (−0.255, 0.186) | −0.042 (−0.266, 0.182) | −0.035 (−0.256, 0.187) |
| Late-onset PPD | −0.254 (−0.592, 0.084) | −0.270 (−0.608, 0.069) | −0.144 (−0.476, 0.187) |
| **At 18 months** | | | |
| Early-onset PPD | 0.080 (−0.126, 0.286) | 0.102 (−0.107, 0.310) | 0.104 (−0.103, 0.310) |
| Late-onset PPD | −0.409* (−0.711, −0.108) | −0.401* (−0.704, −0.099) | −0.346* (−0.646, −0.045) |
| **At 24 months** | | | |
| Early-onset PPD | 0.033 (−0.178, 0.244) | 0.074 (−0.139, 0.288) | 0.081 (−0.129, 0.291) |
| Late-onset PPD | −0.614* (−0.933, −0.296) | −0.601* (−0.920, −0.282) | −0.558* (−0.870, −0.246) |
| **At 32 months** | | | |
| Early-onset PPD | −0.002 (−0.215, 0.211) | 0.055 (−0.159, 0.269) | 0.055 (−0.155, 0.264) |
| Late-onset PPD | −0.519* (−0.844, −0.193) | −0.503* (−0.827, −0.179) | −0.468* (−0.782, −0.154) |
| **At 40 months** | | | |
| Early-onset PPD | −0.114 (−0.334, 0.151) | −0.107 (−0.329, 0.115) | −0.115 (−0.327, 0.096) |
| Late-onset PPD | −0.567* (−0.895, −0.240) | −0.569* (−0.898, −0.240) | −0.549* (−0.861, −0.237) |

**Notes.**
*Statistically significant association ($p < .05$).
PPD, postpartum depression.

participants were included in the analyses (Table 1). The relevance of race was not analyzed because only six non-Japanese children were included (one Han Chinese, two Caucasians, two Brazilians, one Peruvian).

On both crude and adjusted linear regression analyses (Table 2), late-onset maternal PPD (occurring at 5–12 weeks postpartum) was significantly associated with reduced expressive language scores at 18 months or later. The estimated reduction in expressive language scores associated with late-onset PPD exposure was approximately 0.3 standard deviation (sd) at 18 months, and 0.5–0.6 sd at 24 months and later, whereas no similar significant reduction was observed in infants exposed to early-onset maternal PPD (Fig. 2).

Adjusted growth curve analysis confirmed the association of PPD with the changes in expressive language scores over the course of the six time points evaluated (10–40 months of age). We found a significant interaction of PPD status with time point (months of age) but not with square of time point ([time point]$^2$; Table 3). Therefore, while retaining the interaction term between PPD status and time point in the model, we re-evaluated the association between PPD exposure and expressive language scores. We found no significant association at the intercept (i.e., expressive language scores at 10 months of age) for either early-onset PPD (difference in sd: 0.163; $p = .15$) or late-onset PPD (difference in sd:
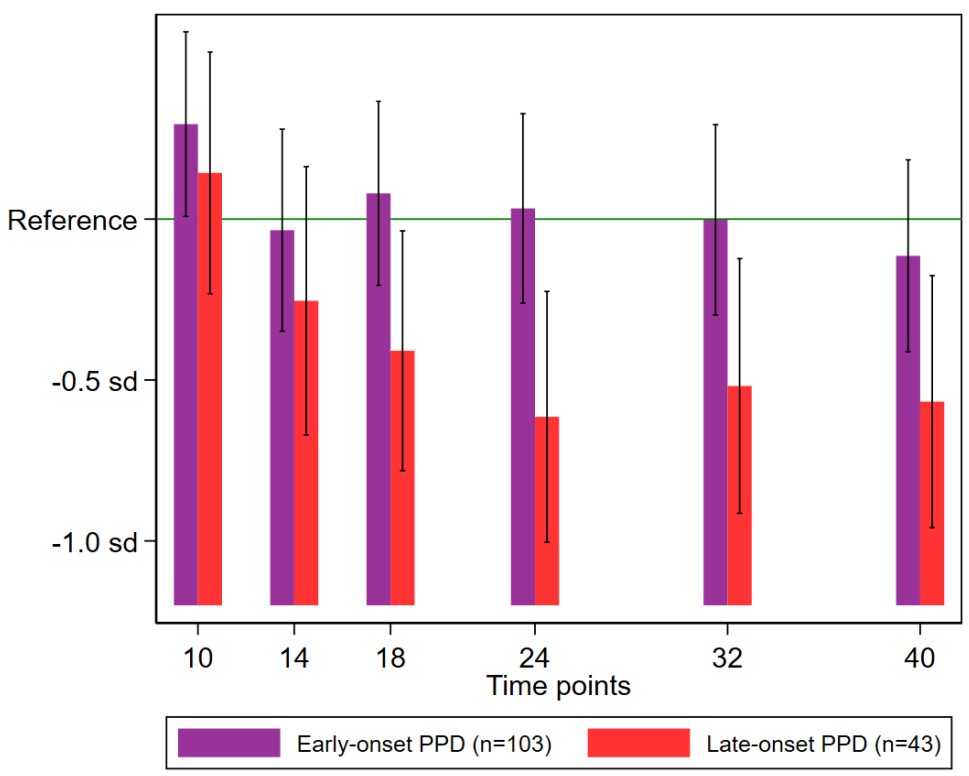

**Figure 2 Expressive language scores among neonates stratified according to exposure to early- or late-onset PPD.** The mean score in the no PPD group was considered as reference. Deviations from the reference value are expressed relative to the sd of the distribution of scores in the no PPD group. The bars (with error bars indicating 95% confidence intervals) correspond to sample-specific mean scores at 10, 14, 18, 24, 32, and 40 months of age, estimated using linear regression adjusted for maternal history of mood/anxiety disorders, infant sex, birth order, twin birth, birthweight, gestational age at birth (in weeks), duration of breastfeeding (in months), maternal and paternal age at the infant's birth (in years), and maternal and paternal education level (in years). PPD, postpartum depression; sd, standard deviation.

0.090; $p = .62$). However, there was a significant decreasing pattern of expressive language scores with increasing age among infants exposed to late-onset PPD (sd change per month, $-0.019$; $p = .002$), though not among those exposed to early-onset maternal PPD (sd change per month, $-0.006$; $p = .14$) (Fig. 3).

## DISCUSSION

To our knowledge, this population-representative birth cohort study is the first investigation to demonstrate that exposure to late-onset maternal PPD is associated with a delay in neurodevelopment, reflected by reduced expressive language scores at multiple time points through infancy and early childhood. Specifically, infants exposed to late-onset maternal PPD exhibited a monotonic and significant decline in expressive language scores, with values up to 0.6 sd below the reference. Such a large and sustained influence of late-onset maternal PPD on child development should be addressed in clinical contexts. On the other
**Table 3 Interaction between PPD and time point (months of age).** Growth curve modelling was conducted using early-onset PPD, late-onset PPD, time points, and covariates in the fixed part, and time point in the random part. Two sets of interaction terms were also entered into the fixed part: (i) model with PPD vs. time point interaction; and (ii) model with PPD × time point and PPD × (time point)$^2$ interaction. The coefficient for each term represents the change in the sd of the expression language score relative to the no-PPD mean estimate. Only the first model to be statistically supported.

| | Change in sd | *p*-value |
|---|---|---|
| Model in which PPD and time points interact | | |
| Early-onset PPD | 0.1634 | .15 |
| Late-onset PPD | 0.0903 | .62 |
| Early-onset PPD × time points | −0.0058 | .14 |
| Late-onset PPD × time points | −0.0187 | .002 |
| Model in which PPD and (time points)$^2$ interact | | |
| Early-onset PPD | −0.0501 | .98 |
| Late-onset PPD | 0.6518 | .09 |
| Early-onset PPD × time points | 0.0108 | .58 |
| Late-onset PPD × time points | −0.0740 | .04 |
| Early-onset PPD × (time points)$^2$ | −0.0033 | .38 |
| Late-onset PPD × (time points)$^2$ | 0.0011 | .11 |

**Notes.**

PPD, postpartum depression.

hand, no such association was observed for early-onset maternal PPD, which represents an encouraging finding.

*Quevedo et al. (2011)* reported that infants of mothers with PPD at 1–3 months postpartum had delayed language development at 12 months of age. Similarly, *Koutra et al. (2013)* reported that children of mothers with PPD at 2 months postpartum had reduced cognitive scores on the Bayley scale at 18 months of age. Our findings agree with these previous observations despite using different methods and definitions. On the other hand, *Murray (1992)* reported that children of mothers with PPD had no reduction in expressive language scores on the Reynell scale at 6 and 12 months of age. Similarly, *Cornish et al. (2005)* found that infants of mothers with PPD showed no significant reduction in expressive language scores on the Bayley scale at 4 and 12 months of age. Our findings support both the observations of *Koutra et al. (2013)* and those of studies with negative findings by *Murray (1992)* and *Cornish et al. (2005)*. It is important to keep in mind that expressive language and communication skills develop dramatically starting at about 1 year of age (*Reilly et al., 2006*), suggesting that any delay in expressive language development is more likely to be observed at around or after this age.

The definition of PPD onset may also account for some of the discrepancies in the reported findings. Studies that did not find an association between maternal PPD and the child's expressive language development evaluated the incidence of PPD at 4 months after childbirth or even later. Our findings highlight the relevance of adequate screening for PPD especially at 1–3 months postpartum, as such late-onset PPD has a critical influence on infant neurodevelopment in terms of expressive language development. To our knowledge, no such clarification has been made to date, and evidence in the literature

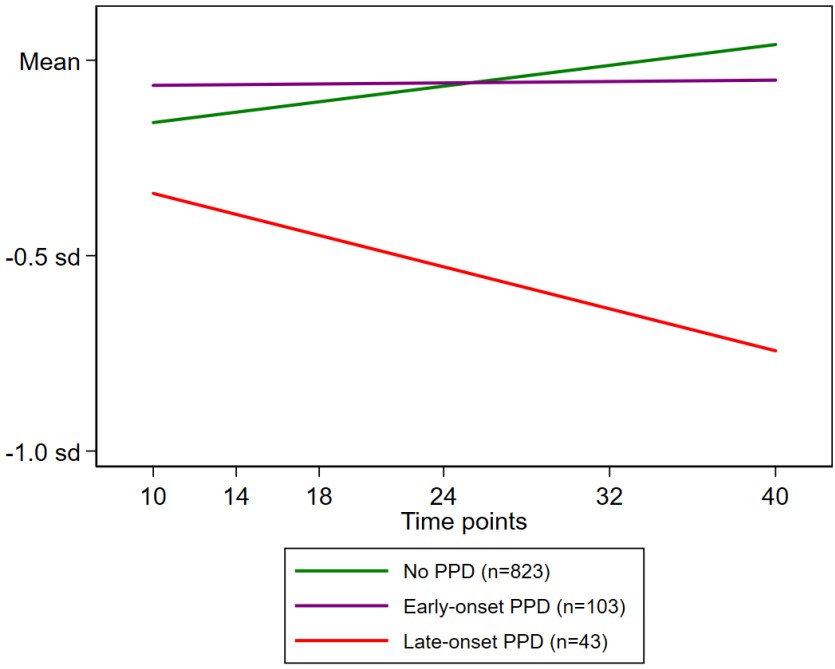

**Figure 3  Time evolution of expressive language scores during infancy according to exposure to maternal PPD.** Mean estimate trends for no PPD (green line), early-onset PPD (purple line), and late-onset PPD (red line) are shown. An interaction term between maternal PPD exposure and infant age (in months), which was statistically significant, was entered into the estimating model. Deviations are expressed relative to the sd of the distribution of normal scores. Mean stands for the normal score. The sample-specific means at 10, 14, 18, 24, 32, and 40 months of age were estimated using growth curve modelling adjusted for maternal history of mood/anxiety disorders, infant sex, birth order, twin birth, birthweight, gestational age at birth (in weeks), duration of breastfeeding (in months), maternal and paternal age at the infant's birth (in years), and maternal and paternal education level (in years).

remains conflicting. In this study, we confirmed that there were no differences of severity and the timing of PPD assessment between early-onset and late-onset PPD (see Table 1 and Table S1). Thus, further studies should consider that the timing of the onset of maternal PPD may be one of the more influential factors impacting child development. On the other hand, it is important to note the differences in assessment duration for PPD (i.e., early-onset was for 1 month and late-onset was for 2 months). Indeed, late-onset PPD may have a lasting effect on child development because of its longer assessment duration.

The contribution of external factors must also be considered. Maternal history of mood disorders is an established risk factor for PPD at any time after childbirth (*Milgrom et al., 2008*; *Mori et al., 2011*; *Silverman et al., 2017*). Maternal anxiety is also a known risk factor for PPD (*Milgrom et al., 2008*). Previous research shows that maternal history of mood disorders, irrespective of its onset, is associated with some general delays in child neurodevelopment in general (*Cycyk, Bitetti & Hammer, 2015*; *Ibanez et al., 2015*). Of note in the current study, the regression coefficients did not diminish after adjusting for maternal history of mood/anxiety disorders (crude model vs. model 1 in Table 2). Furthermore, adjustment for such maternal history did not affect the statistical significance

of the association of late-onset maternal PPD with expressive language scores at 18, 24, 32, and 40 months of age. These findings indicate that the association between maternal PPD and infant language development cannot be accounted for by maternal history of mood/anxiety disorders.

Another external factor to consider is the family's parenting style. Maternal PPD may influence maternal parenting style, resulting in reduced maternal interaction with the infant as early as the first three months postnatal (*Cooper & Murray, 1998*; *Tomlinson, Cooper & Murray, 2005*; *Field, 2010*). PPD may even cause unresponsiveness from mothers (*Cooper & Murray, 1998*) and insecure attachment among infants (*Tomlinson, Cooper & Murray, 2005*). While maternal responsiveness plays a role in child language development (*Tamis-LeMonda, Bornstein & Baumwell, 2001*; *Paavola, Kunnari & Moilanen, 2005*), it is difficult to establish the optimal maternal responsiveness levels to enhance language development. For example, maternal responsiveness to child speech appears to be predictive of enhanced language development, whereas no such benefit has been demonstrated for maternal responsiveness to child exploration. Similarly, maternal responsiveness to the infant is predictive of infant's receptive language skills at 12 months of age, but not of their expressive language skill (*Paavola, Kunnari & Moilanen, 2005*).

In fact, many early studies that postulated the association between maternal PPD with unresponsive parenting styles did not control for parental age, social adversity, or socioeconomic status (*Lovejoy et al., 2000*; *Reilly et al., 2010*; *Ukoumunne et al., 2012*). Social factors such as parental social adversity and lower education levels can result in delayed language development (*Ukoumunne et al., 2012*), which is reflected both in receptive and expressive language skills. Indeed, these social effects are more evident in early years, but is more difficult to track by 4 years of age (*Reilly et al., 2010*). Therefore, it is unlikely that reduced maternal responsiveness would fully explain the association we found between late-onset maternal PPD and lower expressive language scores at 10–40 months. While we do not have data reflecting specific parenting behaviors, we did confirm that the association between maternal PPD and lower expressive language scores remains significant even after controlling for maternal and paternal age and education levels, suggesting that the association we found is not a mere reflection of poor parenting style.

The influence of breastfeeding practices should also be considered. *Leventakou et al. (2015)* reported that breastfeeding for longer than 6 months was associated with increased scores of receptive and expressive language on the Bayley scale. For mothers affected by PPD, continual breastfeeding practices has been suggested to reduce the severity of PPD symptoms (*Dennis & McQueen, 2007*; *Hahn-Holbrook et al., 2013*; *Lessen & Kavanagh, 2015*). In our analysis, the association between late-onset PPD and reduced expressive language scores at 18 months and later remains significant even after adjusting for breastfeeding duration and other potential confounders. Upon examination of breastfeeding duration and proportion of breastfed infants, we found that the proportion of infants breastfed for more than 6 months did not differ significantly between the early-onset and late-onset PPD (63% vs. 74%; $\chi^2 = 1.74$; $p = .19$). Thus, our data cannot confirm whether breastfeeding practices underlie the association between maternal PPD and expressive language scores.

It is of interest to determine why the association of late-onset PPD, not of early-onset PPD, with expressive language development was only observed after 18 months. At 10 months of age, typically-developing infants are expected to comprehend at least one recognizable word or word approximation; at 14 months of age, they are expected to use single words and combine sounds with gestures to communicate; at 18 months of age, they are expected to use two or three-word phrases reflecting grammatical conventions; and finally, at 24–40 months of age, they are expected to have an increasingly broad knowledge and command of words that can be composed into short sentences to respond formally (*Mullen, 1995*). Intriguingly, language region in the brain is active in neonates as young as 2–3 months of age (*Dehaene-Lambertz, Dehaene & Hertz-Pannier, 2002*). Consequently, external factors within this time period such as late-onset maternal PPD (with onset at 1–3 months after childbirth) may affect brain regions expected to play a central role in expressive language development at 18 months and later. However, further studies are needed to clarify the underlying mechanism of the long-lasting influence of PPD on brain development soon after birth. One possibility is that the neural basis of language among those with parturients having early-onset PPD might have been less affected than those with parturients having late-onset PPD. Such explanation would lend further support to examining the differential effects of early- and late-onset PPD.

Furthermore, we have to take care of severity and duration of the illness for both groups since, if late-onset PPD were more severe and lasted long, the different outcomes of language development of the two groups of PPD may be accounted for by the severity and/or duration of maternal PPD. Regarding the severity of maternal PPD, we found that the mean of the maximum EPDS scores was higher for the early-onset group compared to the late-onset group (11.7 points at 2 weeks postpartum for early-onset PPD vs. 8.7 points at 5–12 weeks postpartum for late-onset PPD). This is contrary to the putative hypothesis that severe depressive symptoms of late-onset PPD groups might account for larger detrimental effects on the infant's language development. Rather, we can safely assume that the severity of the maternal PPD symptoms has little to do with the language outcomes of the child. As for the duration of the maternal PPD, unfortunately, we did not precisely measure it; we cannot completely refute the possibility that longer duration of the depressive symptoms of the late-onset groups may account for the language delay of the offspring. Nevertheless, we measured maternal depressive symptoms of the parturients at 10 months postpartum, indicating that 19% of early-onset PPD and 21% of late-onset PPD remained PPD (i.e., above cutoff levels) at 10 months. This analysis indirectly suggests that the proportion of those having detrimental effect of PPD between the two groups are the same, and that the duration of the PPD does not seem to be longer in late-onset PPD at 10 months of age. Caution should be taken in interpreting these results, however, as we do not have any direct measures for the duration of the PPD.

Few clinical studies have considered the long-standing influence of maternal PPD on the infant's neurodevelopmental outcomes. We found that, although the incidence of late-onset PPD is relatively low (4%), the associated reduction in expressive language scores was substantial and long-lasting, indicating that the clinical implications of this condition are much more serious than previously imagined. Thus, we stress the importance of

monitoring for PPD especially at 1–3 months postpartum to facilitate detection and prompt intervention. These steps would minimize both the immediate and long-lasting negative effects on the child's neurodevelopment.

A key strength of the current study is the relatively large sample obtained from a population-representative birth cohort rather than from a clinical case series. Furthermore, the study design was prospective and the outcome was not known at time of enrolment, which minimized selection bias. Our method for measuring language development was also advantageous in that the evaluation with MSEL was conducted by trained assessors, rather than the caregivers themselves, which minimized information biases. Finally, our dataset that included several variables that were potential confounders for this study (e.g., socioeconomic indices for both parents), allowing us to control for these confounds in our analyses.

The current study also had some limitations, including a relatively small sample size of the late-onset PPD group ($n = 43$, 4%), which precluded the testing of dose–response relationships. While around 10% of data points from the expressive language evaluations were missing at each time point, we determined that the missing data points occurred completely at random. Secondly, PPD was defined based on EPDS scores which, though reliable (*Yoshida et al., 2001*), were originally meant for screening and not for diagnosis. Further, in-depth assessment of depressive symptomatology, such as suicide ideation, and follow-up of the symptoms have not been conducted. Self-reported data, leading to recall bias and social desirability bias, cannot be completely ruled out. To account for these biases, maternal neurocognitive capacity (e.g., IQ) should have been considered (although, note that we have introduced covariates such as maternal education level in our analyses). A single-center study, based on a university hospital, might have also distorted the results. In addition, parturients have free access to maternity clinics in Japan. However, our sample is representative of Japanese parturients. Furthermore, as studies from different countries have reported similar prevalence of PPD (*Halbreich & Karkun, 2006*), as well as PPD risk factors (*Milgrom et al., 2008*; *Mori et al., 2011*), we believe that the generalizability of the findings is not compromised. Unfortunately, we have no data on the duration of maternal PPD; as such, we could not directly assess the impact of the PPD duration on the infant's language development. Finally, the evaluation of language development in the study (i.e., MSEL) is a single, validated measure. While previous studies have used other measurements such as the Bayley Scales of Intellectual Development, the advantage of the MSEL is that it obtains non-verbal and verbal skills separately (*Akshoomoff, 2006*). Expressive language development is guided by the development of non-verbal communication skills, which occurs earlier in life (*Iverson & Goldin-Meadow, 2005*; *Kawai et al., 2017*). Additionally, expressive language skills might reflect interaction with other domains, such as motor skills and receptive language (*Mullen, 1995*). However, in the present study, we focused solely on expressive language skills, because the development of speech, rather than that of comprehension or movement, tends to be more appreciated by parents.

## CONCLUSIONS

In this population-representative birth cohort, maternal PPD emerging at 1–3 months postpartum was a risk factor for delayed expressive language development of the offspring. While this delay is not evident before 18 months of age, the reduction in expressive language skills at and beyond 18 months is substantial and long-lasting, with a reduction of up to 0.6 standard deviations below the reference level. The results of the current study highlight the importance of carefully monitoring for PPD at 1–3 months postpartum in order to facilitate early detection and effective intervention.

## ACKNOWLEDGEMENTS

The authors thank Dr. Tetsuo Kato of the Kato Maternity Clinic, as well as Drs. K Sugihara, M Sugimura, K Takeuchi, K Suzuki, Y Murakami, Y Kohmura, Y Miyabe, K Hirai, Y Nakamura, R Koizumi, H Murakami, Y Kobayashi-Kohmura, K Muramatsu-Kato, Prof. H Itoh, Prof. N Kanayama, and all attending obstetricians of Hamamatsu University School of Medicine, for their full support with participant enrolment in the study. The authors also thank the chief midwife, Ms. Kiyomi Hinoki, and all midwives and staff at the maternity clinic of Hamamatsu University School of Medicine, for their support with participant enrolment. The HBC Study team includes E Higashimoto, N Kodera, A Nakamura, C Nakayasu, Y Nishimura, H Suzuki, Y Kugizaki, Y Suzuki, R Takabayashi, M Honda, H Muraki, M Narumiya, E Sato, Mr. R Nakahara, Drs. D Choi, T Harada, T Horikoshi, A Okumura, M Tsujii, K Wakusawa, Y Kameno, D Kurita, H Kuwabara, K Takebayashi, M Yokokura, T Wakuda, R Asano, T Ismail, K Iwata, Y Iwata, E Kawai, M Kawai, YK Kuroda, K Matsumoto, H Matsuzaki, N Mori, T Mori, K Nakaizumi, K Nakamura, AA Pillai, Y Seno, C Shimmura, S Suda, G Sugihara, T Sugiyama, K Suzuki, K Yamada, S Yamamoto, Y Yoshihara, Y Endoh, K Hirano, and T Suzuki. We would like to thank Editage for English language editing.

### Funding

This study was supported by Grants-in-Aid for Scientific Research (B) [16H05374] (to Tsuchiya). The funders had no role in study design, data collection and analysis, decision to publish, or preparation of the manuscript.

### Grant Disclosures

The following grant information was disclosed by the authors:
Scientific Research (B): 16H05374.

### Competing Interests

The authors declare there are no competing interests.

## Author Contributions

- Sona-Sanae Aoyagi and Kenji J. Tsuchiya conceived and designed the experiments, performed the experiments, analyzed the data, contributed reagents/materials/analysis tools, prepared figures and/or tables, authored or reviewed drafts of the paper, approved the final draft.
- Nori Takei and Yoko Nomura conceived and designed the experiments, contributed reagents/materials/analysis tools, authored or reviewed drafts of the paper, approved the final draft.
- Tomoko Nishimura conceived and designed the experiments, performed the experiments, contributed reagents/materials/analysis tools, authored or reviewed drafts of the paper, approved the final draft.

## Human Ethics

The following information was supplied relating to ethical approvals (i.e., approving body and any reference numbers):

The study protocol was approved by the Ethics Committees of Hamamatsu University School of Medicine and Hamamatsu University Hospital (20-82, 21-114, 22-29, 24-67, 24-237, 25-143, 25-283, E14-062, E14-062-1, E14-062-2, E14-062-3, 17-037).

## Data Availability

The raw code files, generated by Stata ver15.1, are provided as Supplemental File 1 (Supplemental files 1_Aoyagi et al.pdf).

The Ethics Committees of Hamamatsu University School of Medicine and Hamamatsu University Hospital restricted access to the raw data outside of this study team due to the limitations of the participant consent which extends only to publishing the metadata, tabulated and/or analysed data. The raw code is available as the Supplemental File 6.

## Supplemental Information

Supplemental information for this article can be found online at http://dx.doi.org/10.7717/peerj.6566#supplemental-information.

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
