# Peer review of "Association of late-onset postpartum depression of mothers with expressive language development during infancy and early childhood: the HBC study"

_PeerJ, doi:10.7717/peerj.6566_

## Round 0.1 · original submission · Major Revisions

I have also read the paper myself and I agree with all of the reviewers' comments. Some of the reviewers point out that the literature review needs to be more comprehensive (currently it's very brief). Also there is no consideration with regard to severity and duration of maternal postnatal depression as well as demographic factors which may impact the results. There are a number of clarification points with regard to the data analyses.

Reviewer 1 ·

Basic reporting

There needs to be a more comprehensive review of the literature, and in particular consideration given to the effects of severity and duration of maternal postnatal depression.

Experimental design

The severity and duration of disorder are not taken into account and are likely confounded with the characterisation of depression as early or late. Analysis also needs to take account of socio-demographic factors that are typically important moderators of depression effects.

Validity of the findings

likely invalid due to lack of accounting for severity and duration of depression

Reviewer 2 ·

Basic reporting

no comment

Experimental design

The authors should describe the Mullen Scales of Early Learning in detail.

Validity of the findings

no comment

Additional comments

Aoyagi et al. investigated the effects of maternal postpartum depression (PPD) on language development in offspring. The authors found that late-onset (between weeks 5 and 12 postpartum) PPD had a significant effect on expressive language development, and that the effect was not evident in early-onset PPD (within 4 weeks after childbirth). The findings are clinically important, and the manuscript is generally very well-written. However, there are points that need to be addressed.

- The description on the measurement of outcome variable
The authors employed the Mullen Scales of Early Learning (MSEL) to assess expressive language development. An evaluation using MSEL is conducted by assessor(s) but not by a caregiver, which may be one of the strengths of this study. This can be mentioned.
Were the assessors masked to information about previously evaluated data and maternal PPD? These can affect the results, and thus, should be clarified.

- “, including postpartum depression,” in Line 56 should be deleted.
- References should be added to “other studies” in Line 60.

Reviewer 3 ·

Basic reporting

It's clear written, with sufficient background provided
Table and figure were well designed
The content include results relevant to hypothesis

Experimental design

The study was conducted based on a relative large sample size, prospective birth cohort. The methods were clearly described and considering the majority of confounding factors (or at least discussed as limitation)
Using growth curve for time evaluation of expressive language score over different time points is a relative novel idea and easy to understand.

Validity of the findings

The data is statistically sound and controlled
The discussion including the shortage and strength of this study, conclusion are well stated
Only some description of data need to state more clearly.

Additional comments

This study aims to investigate the association between maternal postpartum depression (PPD) and infant’s development of expressive language, focus on the onset of PPD. The manuscript is well written and organized. Some comments are listed:
1. Line 103. The original expressive language scores were converted to Z score. What’s the reference population, this study cohort or general population?
2. Line 138. Does the change in growth refer to change in (expressive language) score?
3. The value in Table 2 refer to mean scores or regression coefficient? Please state more clearly.
4. Please merge line 208 “On the other hand…” with next paragraph (line 213 to 217).

External reviews were received for this submission. These reviews were used by the Editor when they made their decision, and can be downloaded below.

---

## Round 0.2 · Minor Revisions

Thank you for dealing with the reviewers' comments. There are still a couple of minor issues I am afraid.

1) Could you please provide a more detailed description of the MSEL? At the moment, you say "as one of the child composite scale". We need some information such as how many items there are, what dimensions of language the scale covers, and provide a few examples.
2) The manuscript needs to be written in appropriate academic English. There are multiple grammatical and spelling errors as well as errors of language structure. The manuscript needs to be rewritten with improved English before it can be published.

Just so that you are aware: PeerJ offers a fee-based language editing service. If you should require to use this service, please contact PeerJ (at [email protected]) for a quote.

Reviewer 2 ·

Basic reporting

no comment

Experimental design

no comment

Validity of the findings

no comment

Additional comments

The authors have addressed all of the comments.

Reviewer 3 ·

Basic reporting

The whole manuscript were well organized and written in fluent English.
The figures were clearly presented
The results and discussion were relevant to the hypotheses

Experimental design

A longitudinal cohort design with mother infant pair and repeated measure is the strength of this study.
The measurement of exposure and outcome were clearly described

Validity of the findings

Conclusion are well stated

Additional comments

The authors have answered my questions point to point with clear expression.
There are only two minor comments: the introduction part is a little bit to lengthy at current version. Second, the authors add more description about MSEL "as one of the child composite scale". Maybe more information such as how many items? in what dimension of language, or give few examples, will be helpful.

External reviews were received for this submission. These reviews were used by the Editor when they made their decision, and can be downloaded below.

---

## Round 0.3 · Minor Revisions

Thank you for proof-editing the manuscript. I have read it all and I found a number of instances where language editing is needed. I list these below.

line 66 - it should be "neurodevelopment in the offsping"
line 71 - "there is limited studies" should be "there are limited studies"
line 104 - does not last long (delete "in duration")
line 141 - "Expressive language score" - should this be "Expressive Language Scale"?
line 152 - the sentence needs to end at "period". Then start a new sentence with "As language ..."
line 273 - "negative studies" - do you mean "studies with negative findings"?

External reviews were received for this submission. These reviews were used by the Editor when they made their decision, and can be downloaded below.

---

## Round 0.4 · accepted · Accept

Dear authors

Thank you for resubmitting your paper. I am satisfied with your edits and I am happy to recommend the paper for publication.

# External reviews were received for this submission. These reviews were used by the Editor when they made their decision, and can be downloaded below.